# Stratospheric Hydration and Ice Microphysics of a Convective Overshoot Observed during the TPEx Campaign over Sweden

Patrick Konjari<sup>1,2</sup>, Christian Rolf<sup>1</sup>, Martina Krämer<sup>1,2</sup>, Armin Afchine<sup>1</sup>, Nicole Spelten<sup>1</sup>, Irene Bartolome Garcia<sup>1,2</sup>, Annette Miltenberger<sup>2</sup>, Nicolas Emig<sup>2</sup>, Philipp Joppe<sup>3</sup>, Johannes Schneider<sup>3</sup>, Yun Li<sup>4</sup>, Andreas Petzold<sup>4</sup>, Heiko Bozem<sup>2</sup>, and Peter Hoor<sup>2</sup>

Correspondence: Patrick Konjari (p.konjari@fz-juelich.de) and Christian Rolf (c.rolf@fz-juelich.de)

Abstract. This study examines the cloud microphysical properties and trace gas signatures associated with a convective overshooting event observed during the TPEx (TropoPause composition gradients and mixing Experiment) campaign in June 2024 over Sweden. While recent studies have predominantly focused on convective overshootings in sub(tropical) air masses, this particular event took place during a cold air outbreak characterized by low tropopause altitudes of 9 km. The measurements by the microphysical cloud spectrometer reveal that ice particles were transported into the lower stratosphere, with detections extending up to 1.5 km above the tropopause. At this altitude, a pronounced stratospheric ozone concentration of approximately 600 ppbv and a notable tropospheric water vapor concentration of up to 60 ppmv (+30 ppmv) were recorded, the latter being twice as high as background levels at the same height. This substantial injection of tropospheric air and ice particles was linked to gravity wave breaking, and subsequently irreversible mixing near the overshooting top. Forward trajectories indicate that the impact of the overshoot on the lower stratosphere, especially regarding the observed hydration, is relatively short-lived with a life time of several days (at 330 K) to weeks (at 345 K).

<sup>&</sup>lt;sup>1</sup>Forschungszentrum Jülich GmbH, Institute of Climate and Energy Systems 4 – Stratosphere, Jülich, Germany

<sup>&</sup>lt;sup>2</sup>Johannes Gutenberg-Universität Mainz, IPA, Mainz, Germany

<sup>&</sup>lt;sup>3</sup>Johannes Gutenberg-Universität Mainz, MPIC, Mainz, Germany

<sup>&</sup>lt;sup>4</sup>Forschungszentrum Jülich GmbH, Institute of Climate and Energy Systems 3 – Troposphere, Jülich, Germany

#### 1 Introduction

Convection plays an important role in transporting air masses from lower regions altitudes into the upper troposphere (UT), sometimes even penetrating the tropopause and injecting tropospheric air into the lower stratosphere (LS). This phenomenon is known as overshooting convection (OC). It is the fastest mechanism for mixing air between the troposphere and stratosphere—a process referred to as stratosphere-troposphere exchange (STE)—and can result in the irreversible transfer of air masses between the two layers. Various studies have concluded that OC significantly influences the chemical composition of the stratosphere and occurs not only in the tropics but also in the mid-latitudes (e.g., Cooney et al., 2018; Homeyer et al., 2017; Jensen et al., 2024). However, the global frequency of OC, as well as the associated dynamical transport processes and their impact on LS composition, remain poorly understood. This highlights the importance of further studying these events to gain deeper insights into the nature of OC and its potential influence on the global climate. Convective overshoots in particular lead to the injection of water vapor (H<sub>2</sub>O<sub>gas</sub>) and ice into the stratosphere. Their influence on the water vapor H<sub>2</sub>Q<sub>gas</sub> budget is especially important due to the sensitivity of the UT/LS region to changes in water vapor H<sub>2</sub>Q<sub>gas</sub>; even small variations in UT/LS water vapor H<sub>2</sub>Q<sub>gas</sub> have been found to substantially alter the radiative budget (Riese et al., 2012; Banerjee et al., 2019; Gettelman et al., 2011). Previous studies have mostly focused on convective hydration in the tropics, with particular emphasis on the Asian Monsoon (Rolf et al., 2017; Kaykin et al., 2022; Konopka et al., 2022). However, several studies indicate that convection in the extratropics may also play a significant role in STE, particularly in moistening the extratropical transition layer (exTL) (Homeyer et al., 2014; Smith et al., 2017; Gordon et al., 2024) (Homeyer et al., 2014; S , with most case studies centered on convective systems over the U.S. For example, based on in-situ measurements over the U.S. Midwest, Homeyer et al. (2023) observed an increase in water vapor H<sub>2</sub>O<sub>gas</sub> of up to 26 % (1.2 ppmv) at 19.25 km altitude (potential temperature: 463 K) originating from a fresh overshoot. Using both in-situ and satellite (MLS) data, Jensen et al. (2024) found that high anomalies of  $H_2\mathrm{O}_\mathrm{gas}$  (> 8 ppmv) above the cold-point tropopause occur predominantly in the extratropics—particularly over North America—and almost never in the tropics, not even during the Asian monsoon season. Despite the high frequency of strong H<sub>2</sub>O<sub>gas</sub> anomalies over North America, it remains to be assessed how these events influence the stratospheric H<sub>2</sub>O<sub>gas</sub> budget over the long term (i.e., several months or more). Wang et al. (2023) used a highresolution numerical weather prediction model to investigate the lifetime of such H2Ogas plumes. Water injected into the stratospheric overworld—defined by potential temperatures ( $\Theta\Theta$ ) greater than 380 K, a threshold chosen because the 380 K is the lowest isentropic surface lying entirely above the tropopause globally, throughout all seasons, thereby marking the lower boundary of the permanently stratospheric air mas—was mostly retained in the stratosphere. In contrast, in the LS, about 50 % of the H<sub>2</sub>O mass was returned to the UT, driven by convectively induced perturbations, several hours after the overshoot (Wang et al., 2023). Cooney et al. (2018) analyzed radar reflectivity data over the U.S. and estimated that 45 % of overshooting events reached the stratospheric overworld. Based on in situ measurements, Sayres et al. (2024) showed that convection increased the water vapor mixing ratio in parts of the North American Monsoon Anticyclone by as much as 40 % at 380 K in the summer season.

5 Concerning the convective impact on the global stratospheric  $H_2O_2O$  budget, the studies by Ueyama et al. (2023) and

Dauhut and Hohenegger (2022) estimate it to be in the range of 10 %. However, this estimate earries a high degree of uncertainty. A is subject to considerable uncertainty, as these studies primarily account for tropical convection while largely neglecting extratropical events. Another major contributor to this uncertainty is the sensitivity of simulations to the choice of convective and microphysical parameterizations (Lu et al., 2018)(Lu et al., 2018; Pandey et al., 2022). For this reason, in-situ measurements are critically important, as they provide essential data for improving the representation of these key processes. This, in turn, helps to better resolve convection and the associated stratosphere-troposphere exchange (STE), ultimately leading to more accurate assessments of the global impact of overshoots on stratospheric composition. Due to the considerable challenges involved in measuring within or near active convective systems, however, in-situ data on overshoots remain scarce. Most of the existing measurements were taken in the vicinity of convective cores rather than within the cores themselves.

In order to assess the global distribution of OC, satellite data are essential. Liu et al. (2020) used satellite observations from the Global Precipitation Measurement Mission (Hou et al., 2014) to map the global distribution of convection reaching the tropopause within the ±65° latitude band. Despite limitations related to satellite resolution, such data provide valuable insights into the global occurrence of these events. Their results indicate that OC occurs frequently not only in mid-latitudes but also in high-latitude regions over N. America, N. Europe, and boreal Asia. Although these events generally do not penetrate as deeply into the stratosphere (Liu et al., 2020), their occurrence in these regions is notable.

An example of a mid-latitude overshoot outside the U.S. was described by Khordakova et al. (2022), who analyzed balloon-borne measurements from a convective system over Germany. They reported significant increases in stratospheric  $H_2O_{gas}$ —for instance, up to 12 (+7) ppmv at  $\Theta$  = 365 K—during two overshoot events. This highlights the need for further studies to investigate the influence of overshooting convection in over Europe and other extratropical regions outside North America.

Our measurements during the TPEx (TropoPause composition gradients and mixing Experiment) campaign in May and June 2024 provide new evidence of a convective overshoot over southern Sweden (59°N). This event occurred significantly further north than any previously documented case with direct in-situ evidence. What makes this case particularly noteworthy is that it was associated with a polar air mass outbreak, occurring in the absence of subtropical air throughout the troposphere. Nevertheless, convection-driven mixing into the stratosphere was clearly observed, including the presence of ice particles several kilometers above the tropopause.

We analyze this case using in-situ measurements of trace gases (water vapor, ozone, nitrous oxide), cloud microphysical properties, and aerosol distributions, in combination with ECMWF ERA5 reanalysis data and satellite observations of  $H_2O_{\rm gas}$  from the Microwave Limb Sounder (MLS) and model simulations by the ICON (ICOsahedral Nonhydrostatic) modeling framework. Additionally, model output from CLaMS (Chemical Lagrangian Model of the Stratosphere) is used to investigate the transport of air masses injected into the LS and the involved ice microphysical processes. The instruments and models used are described in Section 2. The results are presented in Section 3, and SectionSections 4 and 5 provides a discussion and conclusion of the findings, respectively.

Table 1. Summary of the UT/LS airborne in-situ data sets.

| Instruments                                               | Measurement Quantity                     | Uncertainty                | Reference                                |
|-----------------------------------------------------------|------------------------------------------|----------------------------|------------------------------------------|
| Lyman- $\alpha$ photofragmentfluorescencehygrometer(FISH) | $ m H_2O_{ m gas}$                       | 7 % $\pm$ 0.3 ppmv         | Zöger et al. (1999), Meyer et al. (2015) |
| IAGOS capacitive humidity sensor (ICH)                    | $ m RH_{liq}$                            | $\pm5\%~\mathrm{RH_{liq}}$ | Neis et al. (2015)                       |
| 2BTech ozone monitor                                      | $O_3$                                    | $0.7~\%\pm3~\mathrm{ppbv}$ | Johnson et al. (2014)                    |
| UMAQS                                                     | $N_2O$                                   | 0.08 ppbv                  | Müller et al. (2015)                     |
| NIXE-CAPS                                                 | PSD 0.61-937 $\mu m$ & Ice water content | 20 %                       | Krämer et al. (2016)                     |
| Sky-OPC                                                   | PSD 0.25-3 $\mu$ m                       |                            | Bundke et al. (2015)                     |
| UHSAS                                                     | PSD 0.095-1 μm                           |                            | Moore et al. (2021)                      |

#### 2 Data

#### 2.1 In-situ Measurements

During the TPEx campaign, various in-situ instruments onboard a Learjet measured the chemical and microphysical characteristics of the convective overshoot. This includes trace gas measurements (water vapor, ozone, nitrous oxide) and particle size distributions in the ice and aerosol range. The instruments are summarized in Table 1.

#### 2.1.1 Particle size distribution

In this study, we use particle size distributions (PSD) measured by three different instruments that cover distinct size ranges within the aerosol and ice particle domains. The Novel Ice Experiment (NIXE Krämer et al., 2016) instrument is a modified version of the Cloud, Aerosol and Precipitation Spectrometer (CAPS) (called NIXE-CAPS). It measures particles from 0.61 to 937 μm, thereby encompassing the full ice particle size range (> 3 μm). In the UT/LS, the upper size limit of 937 μm is typically only reached during convective events with strong updrafts; otherwise, NIXE-CAPS covers the entire typical range of cirrus ice particle sizes.

Figure 1. Synoptic overview: (a) ERA5 potential vorticity and geopotential height at 360 K potential temperature level. The red box indicates the area of the research flight, and the red star the location of the convective overshoot. (b) For the location of the convective overshoot, Skew-T diagram from ERA5 data. (c) SEVIRI 10.8  $\mu$ m cloud top brightness temperature at 12:30 UTC. The flight path is indicated by the black line.

- To complement the NIXE-CAPS measurements, the lower size range of the aerosol spectrum is covered by the optical particle counter (Sky-OPC; Bundke et al., 2015) and the Ultra-High-Sensitivity Aerosol Spectrometer (UHSAS; Cai et al., 2008; Kupc et al., 2018), which provide particle PSD in the 0.25–3 μm and 0.095–1000 nm ranges, respectively. Both instruments operate based on the light scattering properties of aerosols. The UHSAS uses a Nd<sup>3+</sup>:YLiF<sub>4</sub> solid-state laser emitting at 1054 nm, while the Sky-OPC employs a Class 3B laser diode emitting at 655 nm.The two instruments and their configuration during TPEx are described in detail in (Joppe et al., 2025) and Bozem et al. (2025; in prep).
  - Because the NIXE-CAPS PSDs has limited significance in the aerosol size range (i.e.,  $

Figure 2. Flight track: (a) Flight path and corresponding ERA5 potential vorticity; (b)-(d) Images taken from the aircraft, with (b) showing the convective system from the distance, (c) the anvil cirrus of a cell from close up, and (d) overshooting convection penetrating the anvil cirrus.

overlapping particle size range covered by both the Sky-OPC and UHSAS, the measurements show good agreement (Bozem et al., 2025; in prep.).

#### 2.1.2 Trace gas measurements

100

105

This study analyzes measurements of water vapor ( $H_2O_{gas2}O_{gas}$ ), ozone ( $O_3$ ), and nitrous oxide ( $N_2O$ ).  $H_2O_{gas}$  is of particular interest in this convective case due to the substantial upward transport of water vapor  $H_2O_{gas}$  during such events.  $N_2O$  and  $O_3$  serve as tracers to differentiate between tropospheric and stratospheric air masses:  $N_2O$ —O remains nearly constant in the troposphere (approximately 340 ppbv in this case) but decreases in the stratosphere, whereas  $O_3$  exhibits the opposite behavior, with typical concentrations around the tropopause near 100 ppbv (Zahn and Brenninkmeijer, 2003). The combined use of stratospheric and tropospheric tracers enables the identification of flight segments influenced by stratosphere-troposphere exchange (STE) processes.

The water vapor  $H_2O_{gas}$  mixing ratio is measured using the Fast In-situ Stratospheric Hygrometer (FISH; Zöger et al., 1999). The instrument detects low water vapor  $H_2O_{gas}$  concentrations typical of the UT/LS with an accuracy of 6–8 % in the range of 1 to 1000 ppmv (Meyer et al., 2015). Measurements are provided with a frequency of 1 second.

Additionally In addition to the  $H_2O_{\rm gas}$  measurements by FISH, a IAGOS capacitive humidity sensor (ICH; Neis et al., 2015) was installed onboard, which provides measurements of relative humidity with respect to liquid ( $RH_{\rm liq}$ ). Measurements are provided every second. However, at low temperatures in the UT/LS, the sensor's response time can extend to several tens of seconds.

he The ICH is equipped with a platinum resistance sensor (Pt100), allowing simultaneous measurement of ambient temperature alongside humidity. The overall uncertainty of the ambient air temperature is ±0.5 K, which includes the error of Pt100 itself and the error arose resulting from the data processing (Berkes et al., 2017; Petzold et al., 2020). The temperature data were used for converting relative humidity into water vapor  $H_2O_{gas}$  mixing ratios and for calculating potential temperature.

Due to an offset in the FISH water vapor measurements caused by a modification  $H_2O_{gas}$  measurements for this specific flight, the ICH was used to correct the offset in the FISH water vapor datafor this specific flight  $H_2O_{gas}$  data. While FISH provides higher temporal resolution and is more effective in capturing rapid fluctuations and filaments of water vapor  $H_2O_{gas}$ , the ICH supports this by ensuring measurement accuracy through offset adjustment. Rolf et al. (2023) has demonstrated that both instruments usually exhibit good agreement in principle in this water vapor  $H_2O_{gas}$  range.

 $N_2O$  is measured using the Quantum Cascade Laser-based spectrometer UMAQS (Müller et al., 2015).  $N_2O$  concentrations are derived from its absorption feature at 1006 cm<sup>-1</sup>, with measurements provided at a frequency of 1 Hz and an uncertainty of 0.08 ppbv.

 $O_3$  is measured using a 2B Technologies ozone monitor (Johnson et al., 2014)(Bozem et al., 2025; in prep.), (Johnson et al., 2014; Bozem et al., 2015; in prep.), (Johnson et al., 2014; Bozem et al., 2016), which exploits ozone's strong absorption at approximately 254 nm. Ambient air is pumped into a detection cell where a low-pressure mercury lamp emits ultraviolet light; a photodiode measures the transmitted light intensity. With its measurement range from 1 ppbv to 100 ppmv, the instrument can capture both high stratospheric (>100 ppbv) as well as lower tropospheric concentrations, with an uncertainty of 0.7% + 3 ppbv.

## 2.2 Satellite measurements by the Microwave Limb Sounder and Meteosat Second Generation

In this study data from the Microwave Limb Sounder (MLS) launched on NASA'S Aura Mission in 2004 (Schoeberl et al., 2006) are utilized. Water vapor Level 1.5 data (Version 5) are taken from Goddard Spaceflight Center DAAC (http://mls.jpl.nasa.gov).

MLS scans the microwave emission from the earth atmosphere at its limb, from surface height to 90 km. With its near-polar and sun-synchronous orbit, MLS covers the region from 82° S to 82° N. The resolution is 200 to 300 km horizontally and 2.5 km (in 316–215 hPa) vertically. The accuracy between 316 and 147 hPa is better than 25 %. The precision decreases in regions with increasing water vapor gradients, due to resolution limitations. Read et al. (2007) found a precision of 65 % in at 316 hPa, and still 25 % in at 147 hPa. For the characterization of the sampled and overflown convective system, Level 1.5 reflectivity observations by MSG (Meteosat Second Generation) SEVIRI (Spinning Enhanced Visible Infra-Red Imager) at 10.8 μm (SEVIRI channel 9) are utilized (Bedka, 2011). The data were downloaded from the Eumetsat data platform (https://data.eumetsat.int).

# 2.3 Model data

120

130

#### 2.3.1 ECMWF ERA5 reanalysis

The ERA5 reanalysis dataset (Hersbach et al., 2020), provided by the European Centre for Medium-Range Weather Forecasts (ECMWF), offers hourly meteorological parameters from 1959 to the present. These data are available at a horizontal resolution

Figure 3. In-situ measurement overview: During the 12 June 2024 flight, trace gas measurements; in dashed eyan-magenta 100 ppbv the  $O_3$  line (a), pressure, potential temperature and potential vorticity (ERA5); ; blue dashed line indicates the 2 PVU level (b), potential vorticity cross section and flight path (white dashed ((c)), and measurements of the particle size distribution by NIXE-CAPS (3 - 937  $\mu$ m), Sky-OPC (0.25 - 3  $\mu$ m) and UHSAS (0.095 - 0.22  $\mu$ m). The grey areas indicate the area north of 57° N, influenced by the convection.

of approximately 30 km, with 137 vertical model levels extending up to an altitude corresponding to 0.1 hPa. In this study, hourly ERA5 data are used at a horizontal longitude-latitude resolution of  $0.25^{\circ} \times 0.25^{\circ}$ , on hourly temporal resolution. Along the aircraft flight path, the relative position to the first and, when applicable, second WMO thermal tropopause, as well as

Figure 4. Zoom into the in-situ measurements marked as gray areas in Figure 3. (a-c, f-h) show the trace gas measurements, thermodynamic variables, and the particle size distributions, respectively. In (d,i), ice microphysical properties (IWC and  $R_{ice}$ , as well as  $RH_{ice}$  are provided. The total number of particles for three aerosol size ranges is shown in (e,j).

the equivalent latitude calculated from potential vorticity fields, are interpolated using the CLaMS model toolset (described in Section 2.3.3).

# 2.3.2 ICON

We use the ICOsahedral Nonhydrostatic (ICON) modeling framework (version 2024.7, https://www.icon-model.org/) to place the in situ measurements in a larger context. The simulation starts on the 12.06.2024 at 00:00 UTC and finishes at 15:00 UTC of the same day, covering the initiation of the convection and its development to the mature stage.

ICON allows nested simulations, i.e., a global domain with embedded domains of higher spatial resolution (nests) that can be

**Figure 5. Observations versus potential temperature.** (a) IWC, (b) H<sub>2</sub>O<sub>gas</sub>, (c) O<sub>3</sub>, and (d) N<sub>2</sub>O along the flight track, as function of latitude and potential temperature. The labeling colored boxes indicate in (a) and corresponding labels in (b) denote different filaments of convectively influenced air encountered along the flight pathin panels (b-d). The red dotted line marks the lapse rate tropopause, and the black dotted line in (c) the 100 ppbv O<sub>3</sub> threshold.

coupled in both directions, so called two-way nesting. This means that the coarser parent nest provides the boundary forcing to the higher resolution child nest and the solution of the child nest is nudged into to the parent simulation every time step (Zängl et al., 2022). The set up of the simulation includes a global domain (R03B07 grid with effective grid spacing of  $\frac{13\text{km}}{13\text{km}}$ ) and two nests centered in the region of the observations. The two nests expand from 0-24°E (2-22°E) and 50-66°N (52-64°N). Thereby, the simulation domain includes part of the flight path and covers the region where convection was observed. The innermost domain has a horizontal resolution of  $\approx 3.3$  km (R03B09) with a vertical resolution of 150 m between  $\approx 2$  km and 14 km with a progressively decreasing vertical resolution up to model top at 40 km.




In the global domain and in the first nest, deep convection is parameterized using the Tiedtke-Bechtold convection scheme (Tiedtke, 1989; Bechtold et al., 2008). For the second nest, only shallow convection is parameterized with deep convection being the result of the model dynamics. Non-orographic gravity wave drag, sub-grid scale orographic drag (Lott and Miller, 1997; Orr et al., 2010) and turbulence are parameterized using the standard ICON schemes in all domains. Radiation is treated with the ecRad radiation scheme (Hogan and Bozzo, 2018; implementation in ICON: Rieger et al., 2019). The simulation represents cloud processes with the double-moment scheme by Seifert and Beheng (2006). Recently, Lüttmer et al. (2024) extended their microphysics scheme by additional hydrometer classes, that allow to distinguish ice particles formed for the differentiation of ice particles generated by different formation processes (e.g. immersion freezing, homogeneous freezing, secondary ice production). The used This set up allows for a first analysis on the studied convective case and motivates a future higher resolution simulation to analyze the structure and development of the overshoot in further detail.

#### 2.3.3 **CLaMS**

The Chemical Lagrangian Model of the Stratosphere (CLaMS) is a global, three-dimensional chemical transport model (CTM) (McKenna et al., 2002). The CLaMS trajectory module is based on Lagrangian parcel tracking driven by ERA5 wind and diabatic tendencies for the vertical movement. For the calculation of ice microphysical properties a double moment bulk microphysics scheme (Spichtinger and Gierens, 2009) is utilized along the trajectories which includes schemes for homogeneous and heterogeneous ice nucleation, depositional growth and sublimation, aggregation and sedimentation.

Figure 6. Tracer-tracer correlation. The correlations are shown for  $H_2O_{\rm gas}$  with  $O_3$  (a) and  $N_2O$  (b). The different colors marks the regimes defined in Figure 5.

#### 180 3 Results

### 3.1 Synoptic situation during the flight

A strong trough, associated with The overshoot area lies within a strong trough that is bringing cold air masses of polar origin, lies within the overshoot area and brings unusually cold air masses for this season to the UT and LS, as indicated by the magenta arrow in Figure 1a. As a result, air masses at 360 K are already located deep within the stratosphere, with PV values ranging from 3 to 5 potential vorticity units (PVU), and lapse-rate tropopause heights of approximately 300 hPa (around 9 km), as shown in the vertical profile over the overshoot region in Figure 1b.

The tropospheric air masses can be traced back to the Atlantic sector (40–50°N), with moist conditions due to warm conveyor

belt uplift and subsequent outflow into the overshoot region. Compared to the colder polar air masses in the UT/LS region, these relatively milder tropospheric air masses, combined with a superadiabatic lapse rate near the surface, create favorable conditions for convective instability in the overshoot area.

The flight path is shown in Figure 1c (black line), and the flight path as a function of potential temperature is presented in Figure 2a. The overshooting convection occurred at the northernmost part of the flight path, between  $58^{\circ}$  and  $59^{\circ}$ N. Figure 2b displays the convective system, which had a multi-cellular structure, as seen in the SEVIRI 10.8  $\mu$ m cloud-top brightness temperatures in Figure 1c. Over the area of convection, the aircraft flew at four pressure levels: 330, 295, 245, and 210 hPa—thus covering the UT (PV < 2 PVU; see Figure 2a), the tropopause region, and parts of the LS, where PV values reached up to 9 PVU.

## 3.2 Cloud, aerosol and trace gas in-situ measurements of the overshoot





The measurements along the flight track are summarized in Figure 3, together with interpolated ERA5 data. The aircraft first flew through the convection from 11:40 to 11:55, at two flight levels (330 and 295 hPa; referred to as lvl1 and lvl2, respectively), and then again at two higher levels (245 and 210 hPa; referred to as lvl3 and lvl4) from 12:40 to 13:00 UTC (greyish areas in Figure 3). For these two periods, Figure 4 shows a more detailed evolution of the quantities within the convection.

Around 11:40, the aircraft flew through the anvil cirrus situated just at the tropopause (see also Figure 2c), which is characterized by the presence of large ice particles, up to 500  $\mu$ m in size. This The occurrence of such large particles suggests that the ice particles have a liquid origin, meaning they consist of droplets that froze after being transported upwards (Krämer et al., 2020).

The aerosol concentration slightly increases across all size ranges (Figure 4e), while the concentrations of the trace gases remain relatively unchanged to before.  $O_3$  and  $N_2O$  levels indicate that the flight is at or near tropopause height, with  $O_3$  values ranging from 90 to 105 ppbv and  $N_2O$  at 337 ppbv. The high  $H_2O_{\rm gas}$  concentration, around 150 ppmv, can be attributed to the relatively high tropopause temperature of approximately -55°C and the origin of the air mass, which comes from previously uplifted air over the Atlantic in a warm conveyor belt (see Section 3.1).

As the aircraft enters the convective core, the ice water content (IWC) increased to approximately 1000 ppmv. The highest IWC values peak at around 1450 ppmv. The air was strongly supersaturated, with relative humidity with respect to ice (RH<sub>ice</sub>) ranging from 120 to 160 %. The cloud contains a large number of small particles, as well as very large particles up to the upper detection limit of 0.9 mm. The high particle concentrations are a result of the freezing of previously liquid particles (large particles) but may also stem from additional homogeneous freezing (small particles) after the consumption of all heterogeneous ice nucleation particles, combined with high updrafts and the resulting high supersaturation (Krämer et al., 2016). The significant aerosol transport is evident from the size distributions, with total number concentrations increasing by a factor of 10<sup>2</sup> (<0.25 μm) to 10<sup>3</sup> (>0.25 μm).

 $H_2O_{gas}$  increases to around 200 ppmv, with a peak near 250 ppmv.  $O_3$  values are around 50 ppbv, compared to approximately 100 ppbv (magenta dashed line) before entering the convection.  $N_2O$  predominantly shows tropospheric values; however, strong fluctuations suggest possible mixing of stratospheric air from above.

As the aircraft ascends to Ivl2 (295 hPa) at 11:45, O<sub>3</sub> and N<sub>2</sub>O show strong, anti-correlated fluctuations (50 - 140 ppbv and



Further measurements about the vertical structure, as well as measurements of the vertical wind speed would

Qu et al. (2020) provide a detailed description of different overshoot-related gravity wave breaking mechanisms and related STE.

As the aircraft leaves the convective core at 11:47, O<sub>3</sub> increases to 140 - 150 ppbv while N<sub>2</sub>O decreases to 335 ppbv which

Figure 7. ICON simulations:  $H_2O_{gas}$  (a, c, e) and IWC (b, d, f) on three different pressure levels are shown. The flight path is shown as black solid line. Magenta circled areas indicate regions below the thermal tropopause, all other areas are located above tropopause. The red dashed line marks the location of the cross section shown in Fig. A1 in the Appendix.

indicates that this segment is located slightly above the tropopause level. At the same time, ice particles are still present but concentrations and IWC are very low compared to the convective core (IWC of  $10^{-1}$  to  $10^{1}$  ppmv). The occurrence of ice

Figure 8. Comparison of in-situ measurements with MLS, ERA5 and ICON. (a) MLS observations (circles) of water vapor at the 260 hPa level, taken around 12 UTC on the day of the convection event. Overlaid are The thermal tropopause pressures pressure from ERA5 reanalysis are indicated by the background color. Red dashed lines represent 24-hour forward trajectories. Grey shaded areas indicate where cloud-top temperatures were below –40 °C during the MLS overpass. Magenta contours mark cloud tops that reached or potentially exceeded the tropopause. (b) Vertical pressure profiles of water vapor from multiple sources (gray in-situ, green and blue orange MLS, orange blue ICON and red ERA5). Two MLS averaged profiles are shown: one for the region between 59° and 61° N close to the convection (MLS-2), and another for four profiles located to the north and south of this zone (MLS-1); corresponding mean tropopause levels are indicated by the dotted lines. The dashed line and the shaded area indicate the median and range of MLS observation, respectively. For the ICON model, averaged over the white box region shown in (a), the median profile (dashed line) is plotted along with confidence intervals (shaded) representing the 5–95 %, 1–99 %, and 0.1–99.9 % ranges. ERA5 mean profile is shown for the same area as for ICON. In-situ measurements are also included, with different colors corresponding to specific areas defined in Figure 5.

particles above the overshoot top is referred to as above-anvil cirrus and is a distinct feature that often occurs during OC events. 

It was These features were found to occur through the injection of ice particles related to gravity wave breaking at or close to the overshooting top (Homeyer et al., 2017).

 ${
m H_2O_{gas}}$  stays at about the same amount ( $\approx$  100 ppmv) as before, with RH<sub>ice</sub> of 120 %. This can be explained by the sublimation especially of the smallest particles, which is also evident from the ice PSD (>3  $\mu$ m). It The PSD is now characterized by particles in the size range of 30-120  $\mu$ m, likely because of the sublimation of the smaller particles while the large particles (> 120  $\mu$ m) either. The largest particles likely sedimented or didn't reach this level. The small particles might be missing, because small ice particles grow in supersaturation to sizes larger than roughly 20  $\mu$ m. If no new ice particles are nucleated

Figure 9. Forward trajectories. For three different  $\Theta$  levels, an ensemble of forward trajectories originating in the area of overshooting occurrence (58° - 59° N, 11° - 14° E) is calculated using CLaMS. For the median (dashed lines) and the 10-90 % percentile (shaded), the plots show the evolution in coordinates of  $\Theta$  (a),  $\Theta$  difference to thermal tropopause (b) and PVU (c). The 2 PVU level in (c) is indicated by the black line.

(i.e.,  $RH_{ice}$  remains below the heterogeneous/homogeneous freezing thresholds), the size range below 20  $\mu$ m is depleted. This state is referred to as 'matured cirrus' (Krämer et al., 2025). However, it is also possible that the missing of the small particles in the PSD is a result of a detection limit of the NIXE-CAPS for small particles if occurring in small concentrations.

At around 11:53, the air becomes significantly drier (H<sub>2</sub>O<sub>gas</sub> ≈ 60–70 ppmv), which occurs alongside a slight increase in potential temperature, associated with a temperature rise (not shown) from 219 K (11:45) to 224 K. The lower observed temperature within the filament influenced by the overshoot is likely explained by diabatic cooling due to ice particle sublimation. This diabatic effect accounts for the stronger tropospheric trace gas signature within the overshoot at the same pressure level and is generally recognized as a process by which air masses injected into the stratosphere can partly be transformed back into the troposphere (Homeyer et al., 2024). Furthermore, radiative cooling near the cloud top could have contributed to the observed temperature decrease.

After about one hour, at around 12:35, the area of convection is reached sampled again (see Figure 2a), now at lvl3 (235 hPa). Here, the trace gas measurements indicate that the aircraft is situated in air masses with strong stratospheric influence.  $O_3$  values around 550 ppbv and  $N_2O$  values around 315 ppbv indicative for a strong are strongly indicative of stratospheric character, and  $H_2O_{gas}$  is around 30 ppmv. At 12:40, while flying through the overshoot, the trace gas measurements show strong fluctuations ( $O_3$ : 450–650 ppbv;  $H_2O_{gas}$ : 20–60 ppmv). At the same time, filaments containing ice particles are observed, with low IWC of

 $10^{-1}$  to 1 ppmv. The air is strongly subsaturated (RH<sub>ice</sub>: 20–40 %), indicating that the ice particles must have been transported to this location and didn't form there.

The trace gas fluctuations correlate with strong variations in potential temperature, which result from temperature fluctuations of several degrees, while pressure remains nearly constant. Such pronounced fluctuations are explained by intense transport of momentum and energy from the overshoot, typically caused by wave breaking (e.g., Sang et al., 2018; Qu et al., 2020) which leads wave activity. This could potentially lead to cross-isentropic transport of air from the overshoot to higher regions-(Qu et al., 2020) provide a detailed description of different overshoot-related gravity wave breaking mechanisms and related STE, through breaking of the wave (e.g., Sang et al., 2018; Qu et al., 2020).




In order to assess potential cross-isentropic mixing of air, Figure 5 (b) to (d) show the trace gas measurements as a function of Θ. Measurements corresponding to different regimes in (a) are marked with dots in the respective box colours. Measurements outside these boxes are assumed to represent background conditions not notably influenced by the convection (greyish).

For lv13 and lv14 (corresponding to  $\Theta$  of  $\approx$  325 K and 345 K, respectively), measurements are available on the respective potential temperature levels both within and outside the convection, allowing for an estimate of the influence of cross-isentropic transport on local composition. For  $O_3$  and  $N_2O$ , the observed fluctuations can be attributed to variations in the potential temperature field, and both tracers exhibit no significant deviation from the background. In contrast,  $H_2O_{\rm gas}$  reveals filaments of both anomalously moist and drier air masses. Notably, air masses associated with ice particle occurrence (58.25–58.5° N; blue in Figure 5) are even drier than the background, suggesting that turbulent downward mixing from higher layers may contribute to this dry anomaly.

Examining N<sub>2</sub>O, values are also lower than background in this segment—similar to the >58.5° N filament (dark blue), where H<sub>2</sub>O<sub>gas</sub> shows values of up to 60 ppmv, representing an a moisture enhancement of about 30 ppmv compared to the background. Overall, the N<sub>2</sub>O signature indicates potential net downward transport throughout the entire overshoot regime at lvl3, as covered by the measurements. This observation aligns with the findings of (Dauhut et al., 2018) and (Frey et al., 2015) several studied (Gordon and Homeyer, 2022; Dauhut et al., 2018; Frey et al., 2015), who, based on large eddy simulations, showed that drier and warmer air from above intrudes after the collapse of the cold overshoot. This intrusion then facilitates further cross-isentropic mixing between overshoot-related air masses and those originating above, and facilitates sublimation of ice as the airmass is typically drier. However, any indication of gravity wave breaking cannot be confirmed, as this would also produce stronger fluctuations in tracers other than H<sub>2</sub>O<sub>gas</sub>.

The enhanced  $H_2O_{gas}$  observed in parts of the dark blue segment can thus be explained by the injection of ice followed by sublimation-driven moistening—implying that ice, rather than air mass transport, serves as the primary moisture source through the overshoot. Consequently, the potential temperature in the dark blue filament is several degrees lower due to diabatic cooling than in filaments encountered before the overshoot (<58° N). This diabatic descending descend was found to be a prominent feature associated with OC (Homeyer et al., 2024), and could additionally explain deeper stratospheric trace gas signature in parts of the overshoot regime.

At the next higher level (lvl4), corresponding to 345 K, no significant small-scale fluctuations in potential temperature or trace gas concentrations are observed. On a larger scale, however, H<sub>2</sub>O<sub>gas</sub> increases from 8 ppmv (dark blue box) to 15 ppmv (light

blue box), while  $O_3$  remains constant (no valid  $N_2O$  measurements are available at this time). This increase in  $H_2O_{\rm gas}$  may still be linked to the influence of the convection.

At around 13:15 another filament shows a strong HO increase, reaching up to 30 ppmv at 345 K. This filament is located several hundred kilometres away from the overshoot and lies opposite to the direction of the overshoot's forward transport. Back-trajectories indicate that the filament originated over the Atlantic and, prior to that, in polar regions. Thus, it is unlikely to have formed by convective injection from the observed overshoot; a more plausible explanation is alternative transport such as turbulence associated with a warm conveyor belt (WCB) uplift over the Atlantic or the influence of strong vertical wind shear near the jet.

# 305 3.3 Stratospheric moistening from model and satellite perspective




To gain deeper insight into the mixing processes during convection and to validate the simulation against in-situ measurements, we examine the high-resolution simulation ( $\Delta x \approx 3.3$  km) performed with ICON, with the model setting described in Section 2.3.2. Furthermore, ERA5 reanalysis and MLS satellite data are used in the analysis.

For ICON, Figure 7 illustrates the model results for H<sub>2</sub>O<sub>gas</sub> and IWC as horizontal maps at three pressure levels. Examining the structure of the simulated convective system at 290 hPa (Figure 7f), we find very good agreement with the position of the convective cells and the brightness temperature satellite images shown in Figure 2c. At 290 hPa, the edges of the convective cells extend slightly above the thermal tropopause, as indicated by the magenta solid line marking the tropopause level, while the centers of the cells remain within the upper troposphere. This structure is similar to the observed convection, where at constant pressure, the center of the convective cell remained tropospheric, while the anvil cirrus extended into air masses with stratospheric character.

The upward displacement of the tropopause within the convective region is more clearly visible in the cross-section along the line of active convection in Figure A1 shown as red dashed line. Here, the highest tropopause levels are located above regions of strongest convective activity, with updrafts exceeding 4 m/s. The  $H_2O_{\rm gas}$  profile (Figure A1c) reveals intrusions of  $H_2O_{\rm gas}$  into the lower stratosphere at the edges of the convective cells. At 11.5°E, for instance, an intrusion is observed in a region of partly negative  $d\Theta/dz$  (indicative of convective instability), suggesting strong energy and momentum fluxes associated with gravity wave breaking (see Section 3.2). Another intrusion at 13.5° E originates from a cell located farther south and is also marked by a filament exhibiting convective instability (not shown).

Direct intrusions from overshooting convection cannot be identified in this simulation, as the resolution is insufficient to resolve such features. However, previous studies have shown that the dominant pathway for mixing occurs at larger scales via wave breaking (Wang et al., 2023; Qu et al., 2020). Qu et al. (2020) compared Global Environmental Multiscale (GEM) model simulations at different resolutions and found that simulations with 2.5 km and 1 km resolution captured gravity wave breaking-induced mixing well, whereas the 10 km simulation failed to resolve this pathway. The clear indications of gravity wave breaking in the ICON simulation suggest that this process is still resolved even in the 3.3 km simulation, rather that the moistening and ice particle presence in the LS is the result of numerical diffusion.

To assess the representativeness of the ICON simulated stratospheric moistening, the data are compared with the in-situ

measurements. Furthermore, we compare the in-situ measurements to MLS data. Figure 8a shows MLS data at 260 hPa along with the pressure at the thermal tropopause derived from ERA5. Around 12 UTC, the MLS orbit intersects the convective region, located west of the overshoot observed by the research aircraft. In the figure, the magenta line highlights areas where the tropopause is reached —or possibly penetrated— based on ERA5 tropopause data and SEVIRI 10.8  $\mu$ m brightness temperatures. However, keep in mind that the ERA5 tropopause level might be underestimated directly at the convective core as indicated in Figure A1.







Vertical MLS profiles are shown in Figure 8b. The light green shading (MLS-2) represents the range of  $H_2O_{\rm gas}$  from the two profiles at 59°N and 60.5°N close to the convection, while the orange shading (MLS-1) shows the range from four profiles farther north and south. Dotted lines indicate the corresponding mean tropopause levels for each group. The ICON 3.3 km simulation profile is shown for several confidence intervals, 5–95 %, 1–99 %, and 0.1–99.9 %, covering the area highlighted in Figure 8a (white box). In-situ measurements are displayed as colored dots, corresponding to regimes defined in Figure 5.

In the region of convection occurrence convective region, MLS  $H_2O_{\rm gas}$  increases from approximately 5 to 11 ppmv. However, this increase coincides with a slight decrease in tropopause pressure, suggesting greater tropospheric influence at 260 hPa. Comparing in-situ measurements outside the convective region (dark grey; >13°E) with the MLS-2 profiles reveals reasonable agreement at 260 and 215 hPa. These measurements are taken up to 200 km from the convection, with weak and opposing mean winds (<10 m/s), and the convection had only just begun an hour prior. Thus, it is unlikely that these distant air masses had already been affected by convective outflow.

In contrast, in-situ measurements taken at <13°E (light grey) are more consistent with the MLS-1 profile. This suggests that differences between the MLS profiles may not necessarily be driven by convective influence alone. At the tropopause, the MLS profiles is are significantly drier than the in-situ measurements. This underestimation arises because MLS cannot resolve the sharp vertical and horizontal gradients of  $H_2O_{gas}$  in this region. Additionally, its microwave signal is unable to detect the anomalously moist air within and beneath clouds, which are strongly influenced by convection.

When comparing in-situ measurements to the ICON simulation, to the in-situ measurements, the simulation appears to have a dry bias appears in the UT. The We theorized that the simulation underrepresents high supersaturation levels associated with convection—possibly due to model limitations, such as insufficient resolution. However, at and slightly above the tropopause (green dashed line), the ICON simulation captures the contrast between moist convective air masses (The range of the simulation confidence intervals captures the range of the in-situ measurements in dark blue) and the drier surroundings (dark grey)data inside and outside convectively influenced air. Notably, at 280 hPa, some of the moist range exceed values found in the UT. This could be attributed to ice particle sublimation in subsaturated air, as well as higher saturation vapor pressures at elevated temperatures similar as was to that observed in the in-situ measurements. At higher levels, moist filaments observed at 270 and 250 hPa fall within the upper range of the simulation, particularly into the 99.9th percentile.

For ERA5, Figure 8b (red line) shows the mean profile averaged over the same region as for ICON (Figure 8a, white box). Percentiles are not shown for ERA5 due to its relatively coarse resolution of 0.3°. Compared to ICON, ERA5 exhibits significantly higher moisture levels in the LS, suggesting stronger convection-induced stratospheric hydration. Additionally, ERA5 simulates has much higher IWCs above the tropopause (not shown), indicating that the excess water vapor likely

originates from evaporated ice. This may. Unlike ICON, which explicitly simulates convection, ERA5 represents convective effects using a parameterization scheme. The higher moisture levels in ERA5 may thus be attributed to numerical diffusion effects resulting from resolution limitations or from the convection parametrization which might be too strong. Further investigation is needed to determine whether ERA5 systematically overestimates convection-related moistening, and how this might affect the overall water vapor H<sub>2</sub>Q<sub>gas</sub> distribution in the LS.

#### 3.4 CLaMS trajectory and ice model simulations





Lagrangian trajectories are calculated forward for different height levels within the area of convection ( $58^{\circ}$  -  $60^{\circ}$  N,  $11^{\circ}$  -  $14^{\circ}$  E), where satellite data indicate convection exceeding the tropopause level (Figure 8a). An At three different height levels, corresponding to  $\Theta$  of 325, 340 and 355 K, an ensemble of trajectories is calculated which is shown in . The results are shown in Figure 9 for 3 different height levels, with the temporal evolution of median and 10-90 % percentile (dashed line and shaded area, respectively) of  $\Theta$  (a), and  $\Theta$  difference to the thermal tropopause (b) and PV (c).

At 325 K, where a strong moistening was observed, the trajectory median reaches the thermal tropopause level after 2 days. After 9 days, the median PV decreases to below 2 PVU and 90 % of the trajectories are situated below the thermal tropopause, related to diabatic cooling rates. This descent results from a diabatic cooling rate of 1-1.5 K per day. At 340 K, where a slight  $H_2O_{gas}$  increase was still detected in the observations, the median stays above the thermal tropopause even after 10 days (median PV then 6 PVU) but also here some of the trajectories become tropospheric after this time. At both the 325 K and the 340 K level, the trajectories stay in the mid- to high-latitudes after 10 days with a tendency of a northward shift.

The microphysical simulation reveal During the observation of the overshoot at around 12:40, the flight went over ongoing convection, with an visible overshoot observed during the overflight (Fig. 2d). The PSD shows relatively large particles (up to  $200 \mu m$ ), suggesting that smaller ones may have already sublimated under the very subsaturated conditions ( $40 \% RH_{icc}$ ). The microphysical simulation indicates that injected ice crystals sublimate within 3 minutes after completely entrained in the LS about three minutes after being fully entrained into the lower stratosphere by the overshooting convection (not shown). Even sedimentation of ice crystals does not play a significant. Sedimentation plays a negligible role due to their short lifetime in the very sub-saturated environment such dry conditions.

Overall, it is indicated that the injected water vapor H<sub>2</sub>Q<sub>gas</sub> has a rather short life time in the lowermost stratosphere of a few days to weeks. This is in agreement with previous studies that indicated that extratropical overshoots not reaching the stratospheric overworld mostly don't have a long term effect (e.g., Wang et al., 2023) (e.g., Wang et al., 2023; Homeyer and Bowman, 202 . This overshoot case can therefore be categorized as a short term influence of (sub)-polar LMS as a result of large temperature differences between still cold polar regions and heated subtropical regions to the south during late spring/early summer, and the consequent effect on cross-isentropic mixing.

#### 4 Discussion





Trajectory analysis indicates that the impact of convection on the LS is relatively short-lived. Filaments influenced by overshooting convection remained in the LS for several days (at 330 K) to a few weeks (at 345 K). However, it remains crucial to assess the broader influence of such events on LS air masses at higher latitudes during the transition from winter to summer, and to quantify their contribution to the seasonal cycle of extratropical LS water vapor H<sub>2</sub>O<sub>gas</sub> in the northern mid- to high latitudes, particularly during spring and summer (Konjari et al., 2025; Zahn et al., 2014).

Over the United States, overshooting convection is most frequent in May and June (Homeyer and Bowman, 2021). Phoenix and Homeyer (2021) found that springtime convection results in a more substantial moistening of the LS — about 24 % — compared to summer convection, which contributes only to a 7–11 % increase. This enhanced moistening is attributed to higher tropopause temperatures in spring, which allow for higher saturation mixing ratios. Although the TPEx flight occurred later in the year and farther north, the synoptic conditions during the event were more characteristic of U.S. springtime, including high saturation mixing ratios (100 ppmv)—about ten times greater than those observed during low (sub-)tropical cold point temperatures (Phoenix and Homeyer, 2021).

Additionally, the tropopause height during the TPEx event (9.5 km) was even lower than in most springtime overshooting convection cases in the simulations by Phoenix and Homeyer (2021), where the most frequent tropopause heights range from 12 to 14 km. This raises the possibility that the overshooting convection observed during the TPEx flight may be underrepresented in those model simulations, or that the synoptic setup during TPEx differs significantly from typical U.S. overshooting events, located farther south. In the U.S., strong springtime convection is often driven by sharp contrasts between cold Canadian air masses and warm, moist air from the Gulf of Mexico (Neiman and Wakimoto, 1999), which are associated with higher tropopause altitudes. Compared to the TPEx case, these more extreme air mass contrasts in the U.S. may contribute to an even greater potential for overshooting convection.

Satellite observations, however, suggest that overshooting convection, while not always reaching high stratospheric levels, also occurs with similar frequency over boreal Siberia and Canada at 60° N, comparable to major convection regions like the U.S. Midwest (Liu et al., 2020). Further research is needed to identify the conditions under which such high-latitude overshooting events occur—potentially by analyzing reanalysis data and satellite-derived cloud top information. The objective of a future work is TThe present analysis provides the motivation for additional efforts to improve our understanding of the role of extratropical convection in the stratospheric water vapor  $H_2O_{gas}$  budget, particularly in regions such as Europe and the high latitudes, where the impact of overshooting convection is barely studied to this day has received very little attention to date.

## 5 Summary and conclusion

This case study from the TPEx (TropoPause composition gradients and mixing Experiment) campaign, conducted in June 2024 over southern Sweden (59°N), demonstrates that overshooting convection can occur even in the absence of warm, subtropical air masses—specifically during cold air outbreaks in the boreal late spring to early summer. Stratospheric hydration was observed up to 2 km above the thermal tropopause (corresponding to a potential temperature of 345 K), with ice particles

detected up to 1.5 km above the tropopause (330 K). While there is clear evidence of enhanced water vapor in the gas phase (e.g., anomalies of +30 ppmv at 1.5 km above the tropopause), the O<sub>3</sub> and N<sub>2</sub>O profiles do not suggest upward transport of moist tropospheric air into the LS. On the contrary, both tracers indicate the presence of air masses originating from deeper in the stratosphere. This suggests that the observed stratospheric hydration results primarily from the vertical transport and subsequent sublimation of ice particles, rather than from direct injection of moist air. This conclusion is further supported by a local decrease in potential temperature in the filaments enriched with H<sub>2</sub>O<sub>gas</sub>, consistent with latent cooling due to particle sublimation.

Satellite observations from MLS and model simulations with ICON indicate substantial moistening over the convective region. To further evaluate how well models capture convection-driven stratosphere-troposphere exchange (STE), future simulations at higher spatial resolutions are planned. The observed ice microphysical data provide valuable insight into the conditions within the convective system, including the distribution of ice particles and total ice water content. The ice microphysical properties observed during the convective event are summarized in Figure A2. These observations, together with trace gas observations, are crucial for validating model performance and improving the parameterization of convection in numerical models, including the improved representation of overshooting convection and its influence on the chemical composition of the upper troposphere and lower stratosphere, particularly with regard to water vapor.

- . Acknowledgements. The study was funded by the Deutsche Forschungsgemeinschaft (DFG, German Research Foundation) TRR 301 Project-ID 428312742. Parts of this research were conducted using the supercomputer MOGON-NHR and/or advisory services offered by Johannes Gutenberg University Mainz (hpc.uni-mainz.de), which is a member of the AHRP (Alliance for High Performance Computing in Rhineland Palatinate, www.ahrp.info) and the Gauss Alliance e.V. The ICON modelling was funded by the Deutsche Forschungsgemeinschaft (DFG, German Research Foundation) within the GLACIATE project (530241293).
- The authors gratefully acknowledge the computing time granted on the supercomputer MOGON-NHR at Johannes Gutenberg University

  450 Mainz (hpc.uni-mainz.de). The authors acknowledge the team of enviscope GmbH and GFD GmbH for the opportunity to carry out the campaign and the technical support during the campaign.
  - . Data availability. The data will be published during the review process to the Zenodo platform.

440

Author contributions. PK analysed the data, and wrote the manuscript. CR, AA and NS were responsible for the FISH and NIXE-CAPS measurements and helped analyzing the data. IBG and AM provided the ICON simulation and helped with the interpretation of the results.
 JS and PJ provided for the UHSAS and Sky-OPC data. NE and PH provided the ozone and nitrous oxide measurements. YL and AP are responsible for the ICH measurements. HB and PH coordinated the TPEx campaign.

. Competing interests. At least one of the (co-)authors is a member of the editorial board of Atmospheric Chemistry and Physics. The authors have no other competing interests to declare.

Financial support. The article processing charges for this open-access publication were covered by a Research Centre of the 460 Helmholtz Association.

Figure A1. ICON cross-section. Longitudinal cross section (along 58.3° E) through the area of convection. Temperature (a),  $H_2O_{gas}$  (b), IWC (c), vertical wind (d), and potential temperature tendency ( $\Delta\Theta$ ) (e) are displayed. The thermal tropopause derived from the ICON temperature profiles is indicated by the red dashed line, the grey lines indicate different levels of same potential temperature.

Figure A2. Summary of the ice microphysical properties. The median (left) and maximum values (right) of ice number concentration ( $N_{ice}$ ; a and d), effective radius ( $R_{ice}$ ; b and e) and relative humidity w.r.t. ice ( $RH_{ice}$ ; c and f) are shown as a function of IWC and potential temperature.

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
