# Peer review of "Stratospheric Hydration and Ice Microphysics of a Convective Overshoot Observed during the TPEx Campaign over Sweden"

_EGUsphere, 2025_

## Referee Comment (RC3)

**Stratospheric Hydration and Ice Microphysics of a Convective Overshoot Observed during the TPEx Campaign over Sweden**

Patrick Konjari[1,2], Christian Rolf[1], Martina Krämer[1,2], Armin Afchine[1], Nicole Spelten[1], Irene Bartolome Garcia[1,2], Annette Miltenberger[2], Nicolas Emig[2], Philipp Joppe[3], Johannes Schneider[3], Yun Li[4], Andreas Petzold[4], Heiko Bozem[2], and Peter Hoor[2]

[revised manuscript text omitted]
 lvl2 (295 hPa) at 11:45, $O_3$ and $N_2O$ show strong fluctuations (50 - 140 ppbv and 331 - 341 ppbv,

215 respectively), which indicates strong STE due to signatures of UT and LS $O_3$ and $N_2O$ characteristics together with the high IWC of 1000 ppmv. $H_2O_{gas}$ is still very high, with 100–120 ppmv and corresponding ice supersaturation around 120 %. The potential temperature shows only a small increase, except for a local peak when the flight level changes. This can also be seen from Figure 5a, where IWC along the flight track is shown as a function of potential temperature. In general, $\Theta$ gradients are lower within the convection (>58.5°) than outside, and in some segments at lvl2, $\Theta$ is even lower than at lvl1, i.e. static

220 instability occurs (d$\Theta$/dz < 0). Previous studies found that the occurrence of static instability within or close to an overshoot is related to gravity 
[revised manuscript text omitted]

---

## Author Comment (AC1)

**Response to Review 1**

(green: Responses; blue: Included in the updated manuscript)

**General Comments**

• 37: defined by potential temperatures (Θ) greater than 380 K, a threshold chosen because the 380 K is the lowest isentropic surface lying entirely above the tropopause globally, throughout all seasons, thereby marking the lower boundary of the permanently stratospheric air mass.

We included this sentence to the text

• 114: Either explain what the modification was, and with respect to what former setup, or simply state that there is an offset.

We now simply state: 'Due to an offset in the FISH water vapor measurements caused by a contamination for this specific flight, the ICH was used to correct the offset in the FISH water vapor data.'

230: The explanation suggesting that the constant water vapor mixing ratio and RH  $\approx$  120% result from sublimation of small ice particles may need reconsideration. At such levels of supersaturation, even small ice particles are generally expected to grow rather than sublimate, despite the influence of the Kelvin effect. It might be helpful to clarify under which specific conditions sublimation would still be expected at RH > 100%, or to explore alternative explanations for the observed features. Rather than sublimation, the apparent loss of small ice particles under RH≈ 120% could be attributed to preferential growth of larger particles due to a Wegener–Bergeron–Findeisen like process, or to instrumental limitations in detecting the smallest size classes. It may be helpful for the authors to clarify whether such factors have been considered as alternative explanations. While the classical Wegener— Bergeron–Findeisen process involves vapor transfer from liquid to ice, a similar sizeselective growth mechanism may occur among ice particles of different sizes in a supersaturated environment. In such conditions, larger crystals grow faster due to reduced surface curvature effects, while smaller particles may grow more slowly or become depleted through diffusional competition. Clarifying this distinction might help improve the interpretation of the observed changes in the ice PSD.

It is correct that ice crystals do not evaporate under supersaturated conditions. However, a Wegener–Bergeron–Findeisen process is not required here (as the referee also notes, this is a different mechanism). Instead, small ice particles grow in supersaturation to sizes larger than roughly  $20 \, \mu m$ . If no new ice particles are nucleated (i.e.,  $RH_{ice}$  remains below the heterogeneous/homogeneous freezing thresholds), the size range below ~20  $\mu m$  is depleted. This state is referred to as "matured cirrus" (see Krämer et al., 2025, https://doi.org/10.5194/egusphere-2025-669, Figure 1).

Another possibility is indeed that the instrument fails to detect very small particles. The Cloud Aerosol Spectrometer (CAS), which provides measurements for particle size distributions in the <30  $\mu$ m range, may not be sensitive enough to detect very low concentrations. This limitation could lead to an artificial cutoff in the PSD around 30  $\mu$ m.

New statement in the text: 'H2Ogas stays at about the same amount ( $\sim 100$  ppmv) as before, with RHice of  $\sim 120$  %. The PSD is now characterized by particles in the size range of 30-120 µm The largest particles likely sedimented or didn't reach this level. The small particles might be missing, because small ice particles grow in supersaturation to sizes larger than roughly 20 µm. If no new ice particles are nucleated (i.e., RHice remains below the heterogeneous/homogeneous freezing thresholds), the size range below  $\sim 20 \sim \mu$ m is depleted. This state is referred to as 'matured cirrus' (Krämer et al., 2025). However, it is also possible that the missing of the small particles in the PSD is a result of a detection limit of the NIXE-CAPS for small particles if occurring in small concentrations.'

• 236: The explanation invoking diabatic cooling due to ice sublimation is physically sound, but it would be helpful if the authors could quantify the observed temperature decrease in the overshooting filament. Given the latent heat involved, even modest sublimation can cause cooling on the order of 1–3 K depending on the local ice water content. Including this information would help support the proposed interpretation.

Temperatures increased by 5 K, from 219 K to 224 K. From the ClaMS-ice simulation for the overshoot at 12:40, it was found that 20 ppmv of ice leads to approximately 0.5 K of cooling. Given that even higher ice water content likely occurred in this filament (the maximum measured was 25 ppmv, but higher values are realistic, assuming that the 20 ppmv increase in water vapor in the overshoot filament at 12:40 originated largely from ice, as measurements suggest), the observed temperature changes could be explained by sublimation. However, it should also be noted that other processes, such as cloud-top radiative cooling, may have contributed to the observed temperatures.

We added this information to the text: At around 11:53, the air becomes significantly drier ( $H2O_{gas} \sim 60-70$  ppmv), which occurs alongside a slight increase in potential temperature, associated with a temperature rise from 219 K (11:45) to 224 K. The lower observed temperature within the filament influenced by the overshoot is likely explained by diabatic cooling due to ice particle sublimation. This diabatic effect accounts for the stronger tropospheric trace gas signature within the overshoot at the same pressure level and is generally recognized as a process by which air masses injected into the stratosphere can partly be transformed back into the troposphere (Homeyer et al., 2024). Furthermore, radiative cooling near the cloud top could have contributed to the observed temperature decrease.

• 247-251: The observed correlation between trace gas fluctuations and potential temperature is interesting, especially in a context where wave breaking is invoked. Given that wave-induced irreversible mixing tends to reduce such correlations, it would be helpful if the authors could clarify here whether the observed structure reflects an early stage of breaking

with incomplete mixing, or a coherent transport process preceding the mixing. This also in view of the discussion that follows which strengthen the interpretation of mixing.

This is a point that cannot be resolved from our observations. Assessing the occurrence of gravity wave breaking requires vertical measurements—for example, to observe overturning isentropes associated with wave breaking. Along the flight track, vertical velocity measurements would also be helpful; however, no such measurements are available. Therefore, we should be cautious about attributing the observed mixing and corresponding trace gas signatures solely to gravity wave breaking. Evidence against gravity wave breaking being the primary cause is that, except for H2O, the other tracers do not show any fluctuations. This instead suggests that ice particles were transported into the lower stratosphere, because if wave breaking were responsible, fluctuations in potential temperature would be expected for the other tracers as well.

We added this statement: 'However, any indication of gravity wave breaking cannot be confirmed, as this would also produce stronger fluctuations in tracers other than water vapor.'

• 352: The microphysical simulations suggest that ice crystals sublimate within 3 minutes after full entrainment in the LS. However, given the observational evidence of ice particles persisting under subsaturated conditions, it would be helpful if the authors could clarify whether such short sublimation times are consistent with the size range of the observed particles and the inferred degree of subsaturation. Additionally, are the authors confident that the observed particles must have been injected so shortly before detection? Further discussion on the timing and plausibility of such recent injection would be of interest.

We were flying above ongoing convection, with an actual overshoot observed during the overflight (Fig. 2d). The particle size distribution (PSD) shows relatively large particles, up to 200  $\mu$ m. Smaller particles may have already sublimated, which could occur very quickly under such subsaturated conditions.

The environment was highly subsaturated (~40% RHice), and no water vapor enhancements were observed within the filaments containing ice (water vapor increases of up to 30 ppmv occurred outside these areas). If more ice had been present in the filaments at some point, one would expect a corresponding increase in water vapor. However, it is also possible that these ice particles were transported from elsewhere in the lower stratosphere, where sublimation had already taken place.

We added: 'During the observation of the overshoot at around 12:40, the flight went over ongoing convection, with an visible overshoot observed during the overflight (Fig. 2d). The PSD shows relatively large particles (up to 200  $\mu$ m), suggesting that smaller ones may have already sublimated under the very subsaturated conditions (~40 % RHice). The microphysical simulation indicates that injected ice crystals sublimate within about three minutes after being fully entrained into the lower stratosphere by the overshooting convection. Sedimentation plays a negligible role due to their short lifetime in such dry conditions.'

**Specific comments**

**All points below were corrected**

- 110: Typo; "The..."
- 112: "... arising from ..."
- 175: "...masses for the season..."
- 199: Before what?

---

## Author Comment (AC2)

**Response to Review 2**

(green: Responses; blue: Changes in the manuscript)

**General Comments**

• 1. There is some inconsistent use of defined acronyms throughout. In several cases, text alternates between using full text or previously defined acronyms. I recommend acronyms be used uniformly after definition.

Changed accordingly

• 2. There are several instances where parenthetical citations are provided where in-text citations should be used instead. Please carefully review citations and correct as needed. Some examples can be found at lines 91 & 222.

This was because the Bozem et al. paper wasn't published yet when we handed in the paper. In-text citation is now updated.

• 3. Figure labeling needs some improvement. Pressure axes in Figs. 3b, 4b, & 4g list no values, but they should. Not all colors utilized in Figs. 5b-d are defined - namely, what is the difference between light and dark blue? Also, it would be helpful in many instances if illustrations were added to tie into the discussion more. I found it hard to follow interpretations/arguments in some places.

The figures were updated

4. Were vertical velocities measured aboard the aircraft? If so, those would be a valuable
addition to the analysis, particularly to support the arguments with respect to the role of
gravity wave breaking. If not, it should be acknowledged somewhere that such
measurements do not exist.

In the text, we added: 'However, it cannot be determined with certainty whether gravity wave breaking occurred, as additional information on the vertical structure (e.g., of Theta) or vertical wind speed measurements would be required, which were not available during this campaign.'

5. There is a third enhancement in H2O (and attendant changes in other gases) at higher altitudes near 13:15 UTC which is not addressed in the manuscript. It should be acknowledged and an explanation of its possible (or known) source provided as it is clearly evident in Figure 3.

This increase occurs near the convective region but not within the area directly influenced by convection. Moreover, the transport from the convective systems is directed oppositely and therefore cannot explain the observed enhancement. We acknowledged this in the updated version:

'At around 13:15 a second filament shows a strong H2O increase, reaching up to ~30 ppmv at 345 K. This filament is located several hundred kilometres away from the overshoot and lies opposite to the direction of the overshoot's forward transport. Back-trajectories indicate that the filament originated over the Atlantic and, prior to that, in polar regions. Thus, it is unlikely to have formed by convective injection from the observed overshoot; a more plausible explanation is alternative transport such as turbulence associated with a warm conveyor belt (WCB) uplift over the Atlantic or the influence of strong vertical wind shear near the jet.'

**Specific Comments**

• There are a few places where the H2O acronym has an italicized "O". This should be fixed.

Corrected

• The "Homeyer & Bowman (2014)" reference is incorrect. It is missing many authors (should be Homeyer et al. 2014). See http://dx.doi.org/10.1002/2014JD021485.

Corrected

• Line 41: there is another limitation of the Ueyama et al. 2023 & Dauhut and Hohenegger (2022) studies not mentioned here - they primarily capture convection in the tropics, missing much (or nearly all) of the midlatitude overshooting.

Thank you for pointing that out. We acknowledge this in the updated text:

'Concerning the convective impact on the global stratospheric  $H_2O_{gas}$  budget, the studies Ueyama et al. (2023) and Dahut et al. (2022) estimate it to be in the range of 10 %. However, this estimate is subject to considerable uncertainty, as these studies primarily account for tropical convection while largely neglecting extratropical events.'

• Line 48: unnecessary line break; merge with former paragraph.

Merged

• Lines 71-73: this sentence is unnecessary.

I am not sure which sentence you mean since there are several ones in Lines 71-73

Line 107: revise "Additionally" to "In addition"
 Changed accordingly

• Line 110: "he" should be "The"

Corrected

Line 112: "arose" should be "resulting", and "temperature" should be "temperatures"

Corrected

• Line 128: "NASA'S" should be "NASA's"

Corrected

• Lines 133-134: both instances of "in XX hPa" should be "at XX hPa"

Corrected

• Line 141-142: This sentence can be more simply stated as "In this study, hourly ERA5 data are used at a longitude-latitude resolution of 0.25°."

Changed accordingly

• Line 144: please specify the method of interpolation. Linear in space and time??

Interpolation is linear for most variables, exponential in the vertical for water vapor (however, not used in this study)

• Lines 206-208: this sentence seems like too much of a detour and not very relevant to the focus of the analysis (certainly not beyond all that has already been said).

I do not fully agree, as we are also interested in the ice microphysics during convection, even though this aspect is not discussed in detail in the present study.

• Line 221: there are many more studies that can be cited here, especially some of the early modeling studies of Pao Wang and others.

The citation was added.

• Lines 265-267: the role of downward transport is also discussed and shown in a more appropriate analog of midlatitude overshooting simulations — https://doi.org/10.1029/2022JD036713.

The citation was added.

• Line 357: another worthwhile citation to bolster this argument would be the overshoot trajectory analyses of <a href="https://doi.org/10.1029/2021JD034808">https://doi.org/10.1029/2021JD034808</a>.

The citation was added.

---

## Author Response (AR2)

Reply to Editor decision

Dear Aurelien Podglajen,

Thank you for handling the paper as editor. We have gone through all the comments and made the changes accordingly

Best regards,
Patrick Konjari & Christian Rolf